# Estimating soil carbon sequestration potential with mid-IR spectroscopy and explainable machine learning

Yang Hu<sup>1</sup> and Raphael A. Viscarra Rossel<sup>1</sup>

<sup>1</sup>Soil & Landscape Science, School of Molecular & Life Sciences, Faculty of Science & Engineering, Curtin University, GPO Box U1987, Perth WA 6845, Australia.

**Correspondence:** Yang Hu (yang.hu4@postgrad.curtin.edu.au) and Raphael A. Viscarra Rossel (r.viscarra-rossel@curtin.edu.au)

**Abstract.** Soil carbon sequestration refers to the process of capturing atmospheric carbon through plant photosynthesis and storing it in soil as organic carbon. The primary mechanism for carbon sequestration is via organic carbon molecules adsorbing onto mineral surfaces of the soil's fine fraction (clay + silt  $\leq 20 \ \mu m$ ), forming mineral-associated organic carbon (MAOC). Soil has a finite capacity to stabilise and sequester organic carbon, known as carbon saturation capacity, which depends on the proportion of reactive minerals in the soil. The difference between the current MAOC content and the carbon saturation capacity is referred to as the organic carbon saturation deficit ( $C_{def}$ ) or sequestration potential. Fourier-transformed (FTIR) mid-infrared (mid-IR) spectroscopy can simultaneously measure soil properties relevant to carbon stabilisation, organic carbon functional groups, clay and iron-oxide mineralogy and particle size. Therefore, we hypothesise that mid-IR spectroscopy can effectively and accurately estimate  $C_{def}$ . Thus, we aim to (i) develop spectroscopic models to estimate the MAOC and  $C_{def}$ of 482 Australian topsoil samples, (ii) model MAOC and C<sub>def</sub> using mid-IR spectra and an interpretable machine learning, and (ii) interpret the MAOC and C<sub>def</sub> models using the explainable artificial intelligence (AI) algorithm SHapley Additive exPlanations (SHAP). Using frontier line analysis, we fitted a function to the upper envelope of the MAOC vs clay + silt relationship to derive  $C_{def}$ . We recorded mid-IR spectra of the samples and used the regression trees method CUBIST to model MAOC content and  $C_{def}$ . We interpreted these models by examining the regression trees and using SHAP. The models were unbiased and estimated MAOC content with  $R^2$  of 0.86 and RMSE of 2.77 (g/kg soil), and  $C_{def}$  with  $R^2$  of 0.89 and RMSE of 3.72 (g/kg soil). Model interpretation revealed  $C_{def}$  estimates relied on negative interactions with absorptions from organic matter functional groups and positive interactions with absorptions from clay minerals. Our results show that mid-IR spectra can effectively estimate MAOC and soil  $C_{def}$ , offering a rapid and cost-effective method for assessing and monitoring this critical soil function.

#### 20 1 Introduction

Soil organic carbon (C) sequestration refers to the process by which plants capture atmospheric C through photosynthesis and store it in the soil. The United Nations Framework Convention on Climate Change (UNFCCC) has identified soil C sequestration as a critical, nature-based process for withdrawing atmospheric carbon dioxide (CO<sub>2</sub>) (UNFCCC, 2019). Soil organic C sequestration also improves soil health, food and nutritional security, water quality, biodiversity, and elemental

recycling (Lal et al., 2015). Thus, it is crucial to estimate the amount of C soil stores and how much it could store in the future to advance our scientific understanding of C cycling. This understanding will provide the foundation for land managers to develop practices that enhance C sequestration and for policymakers to formulate climate change adaptation strategies. However, estimating the soil C saturation deficit rapidly, cost-effectively, and scientifically remains a challenge.

Soil C from plants begins as particulate organic C (POC). Over time, soil microorganisms consume this POC, while some are broken down into smaller molecules. Some of these molecules are protected from further decomposition through adsorption onto mineral particles, forming mineral-associated organic carbon (MAOC) and providing protection within soil microaggregates (Hassink and Whitmore, 1997; Six et al., 2002; Beare et al., 2014). Soils with higher silt and clay content have a larger mineral surface area and a greater capacity to adsorb and stabilise C. Hassink (1997) found a positive linear relationship between the proportion of clay and silt (particles 

- Zealand soils using pedotransfer functions derived from the quantile regression approach, modelling  $C_{def}$  with mid-IR spectra through partial least squares regression (PLSR), showing good predictability. Similarly, Karunaratne et al. (2024) estimated the  $C_{def}$  of Australian soils using the quantile regression approach and modelled it with mid-IR spectra coupled with PLSR, also achieving good predictability. We did not find other research that estimates  $C_{def}$  using soil spectra. We hypothesise that mid-IR spectra, combined with explainable machine learning, can be used to estimate soil MAOC content and  $C_{def}$  while also providing insights into how the model uses spectral absorption features to identify the soil constituents important for prediction. Thus, we aimed to:
  - 1. Develop spectroscopic models to estimate the MAOC content and the  $C_{def}$  of Australian soils using mid-IR spectra with the regression trees algorithm CUBIST;
  - 2. Interpret these models by analysing the CUBIST rulesets and SHapley Additive exPlanations (SHAP) values to understand how the absorptions of soil organic and inorganic constituents affected model prediction.

#### 2 Methods

70

### 2.1 Soil samples

We used 488 topsoil samples from 275 sites across Australia (Figure 1). The soils were sampled from three depth layers (0–10 cm, 10–20 cm and 20–30 cm). All soil orders from the Australian soil classification were present apart from Anthroposol and Organosol (Teng et al., 2018). Kandosols were the most abundant soil type, followed by Tenosols and Calcarosols, Chromosols and Vertosols, while Rudosols, Dermodols, Kurosols, Ferrosols, and Podosols were present in smaller numbers. Three Hydrosols were excluded from further analysis due to the distinct C storage mechanisms in anoxic soils (Six et al., 2023).

The sampling area spans the main Köppen-Geiger climate zones (Beck et al., 2018), with most of the samples collected from arid hot deserts, with smaller proportions from arid hot steppes and tropical savannahs. Samples were primarily collected from areas with minimal human impact, particularly nature conservation sites, native vegetation grazing lands, and other minimally used areas. Only a small proportion of samples came from production or intensive land use. The vegetation at sampling sites was diverse, encompassing 24 major vegetation groups, with eucalyptus woodlands being the most common type (Commonwealth of Australia, 2020). Most samples were taken from native vegetation or natural bare land, with the rest from non-native vegetation or cleared land (ABARES, 2022).

## 2.2 Soil fractionation

Soil samples were fractionated through physical granulometric separation. The samples were dispersed in deionised water using an ultrasonic probe (Sonics VCX 500 Sonicator, Newtown, Connecticut) with an energy output of 500 J ml<sup>-1</sup> for 200 seconds (Walden et al., 2023). After dispersion, the samples were fractionated using an automated wet sieving apparatus (Analysette 3 Pro, Fritsch GmbH, IdarOberstein, Germany) with 250  $\mu$ m and 50  $\mu$ m sieves. The resulting soils were in three size fractions: macroaggregates (2000–250  $\mu$ m), microaggregates (250–50  $\mu$ m), and the fine fraction ( $\leq$  50  $\mu$ m). The fractionated samples

Figure 1. Location of sampling points.

were then oven-dried at  $60^{\circ}$ C overnight and ground to approximately  $\leq 80~\mu$ m, before the organic C content of each size fraction was measured using an elemental analyser (SoliTOC Cube, Elementar Analysensysteme, Hanau, Germany). The organic C content of the fine fraction, representing MAOC, was recorded in grams per kilogram of whole soil.

### 2.3 mid-IR spectroscopy

The whole soils (sieved to  $\leq 2$  mm) were air-dried before fine grinding to  $\approx 





## 2.4 Frontier lines and calculation of $C_{def}$

The MAOC content of samples displayed a log-normal distribution. We performed a log<sub>e</sub> transformation on the MAOC content and removed three outliers that were more than 1.5 times the interquartile range. We proceeded with the analysis of the remaining 482 samples from 270 sites.

We fitted a monotonically increasing and concave frontier line (Parmeter and Racine, 2013) to the relationship between log(MAOC) and clay + silt content of the samples using the smooth, non-parametric frontier line analysis with the R package SNFA (McKenzie, 2022). We calculated the  $C_{Amax}$  and  $C_{def}$  following the approach described in Viscarra Rossel et al. (2024a). Each point on the frontier line represents the maximum attainable amount of MAOC that soil could store for a particular clay and silt content.

To enable the estimation of uncertainty, we performed 100 non-parametric bootstrap resamples to fit the frontier lines, keeping samples from the same site together during resampling to prevent data leakage. We then averaged all 100 frontier line fits made on the bootstraps. The  $C_{def}$  was calculated as the difference between the estimated mean frontier line and the MAOC content. We also computed the uncertainties of our frontier line estimate by calculating the 95% confidence limits. All values were then back-transformed to their original units for the spectroscopic modelling.

# 2.5 Spectroscopic modelling

The mid-IR spectra were interpolated to  $32 cm^{-1}$  wavenumber intervals to reduce the inherent collinearity of the spectra. Since mid-IR spectra are highly colinear and contain broad absorption features, we interpolated the spectra to  $32 cm^{-1}$  to reduce the redundant information passed into the machine learning model (Deiss et al., 2020). Visual checks confirmed relevant absorption features remained distinguishable at this resolution. We also checked  $C_{def}$  model performance using spectra interpolated to  $8 cm^{-1}$ ,  $16 cm^{-1}$ ,  $24 cm^{-1}$  and  $32 cm^{-1}$  resolutions and found no significant difference between these resolutions. Preprocessing consisted of an initial offset correction, standard normal variate (SNV) transformation, and a final offset correction to address the shift introduced by the SNV transformation. Spectral regions that were either featureless (4000 to 3746 cm<sup>-1</sup>) or containing distracting features from noise and artefacts from water and  $CO_2$  (2370 to 2082 cm<sup>-1</sup>) were removed before modelling.

We modelled the MAOC and the estimated  $C_{def}$  with CUBIST. CUBIST is a rule-based regression tree algorithm (Quinlan et al., 1992; Wang and Witten, 1997). CUBIST creates a tree structure, with branches as a series of "if-then" conditions, then reduced into rules. Each CUBIST rule corresponds to a subset of the data that satisfies the rule's condition. For each rule, a linear regression model is fit to the data using relevant predictors (Kuhn et al., 2012). CUBIST balances accurate predictions and model interpretability through its rule-based structure. CUBIST is tuned by two parameters: committees and neighbours. The number of committees specifies the number of ensembles contributing to the final prediction, with more committees typically improving performance but reducing interpretability, and the number of neighbours specifies how many nearest-neighbours of a sample CUBIST uses to adjust its rule-based predictions. Viscarra Rossel and Webster (2012) described the method for spectroscopic modelling. In our experiments, since our goal was to understand which spectral regions influence predictions and how they





relate to soil properties, we prioritised model interpretability by using only one committee to retain model transparency without the added complexity of ensemble averaging. We optimised the best number of neighbours to be used by testing all numbers from 0 to 9. Model fitting and validation were carried out using 10-fold leave-site-out cross-validation, where the 275 sampling sites were randomly assigned to 10 folds to ensure all samples from the same site and the three depth layers were kept together within the same fold. We assessed the models based on their coefficient of determination (R<sup>2</sup>), Lin's concordance correlation coefficient (CCC) (Lin, 1989) and the root mean squared error (RMSE).

We propagated the uncertainty of the frontier line fitting and the CUBIST modelling. From the 100 frontier line fits made with the bootstraps, we derived the upper and lower 95% confidence intervals (CI) for the frontier line fit and calculated the upper and lower limit of  $C_{def}$ . The upper and lower limits of  $C_{def}$  were also modelled with CUBIST following the same method described above.

## 2.6 Interpretation

To interpret the models, we extracted each CUBIST rule from the MAOC and  $C_{def}$  models to analyse their rule partitioning. For the MAOC model, we examined the distribution of MAOC values within each rule, while for the  $C_{def}$  model, we analysed the distributions of both MAOC and  $C_{def}$  values within each rule. For the linear models in each CUBIST rule, we examined the wavenumber corresponding to specific absorptions of soil constituents and their coefficients. For the  $C_{def}$  model, we took an additional step and calculated the SHAP (SHapley Additive exPlanations) values for each sample for each linear model of the CUBIST rules. The SHAP values are used to explain the outputs of machine learning models. SHAP is based on game theory (Shapley, 1953) and assigns an importance value to each feature (in our case, absorptions at specific wavenumbers) in a model. Features with positive SHAP values had a positive impact on the prediction, while those with negative values had a negative impact. The magnitude measures the strength of the effect.

All statistical analyses were performed using R (R Core Team, 2024).

#### 3 Results

#### 3.1 The maximum attainable MAOC storage, the MAOC deficit and C sequestration potential

Our samples represent a wide geographical area in Australia (Figure 1) with large variations in MAOC content and texture (Table 1). The MAOC content ranges from 0.27 g/kg soil to 50.04 g/kg soil, while silt content ranges from 0.54% to 31.81%, and clay content ranges from 2.34% to 54.25% (Table 1). The frontier line estimates the maximum C that can be stored in their current environments over their range of clay + silt contents for all 482 samples, with their 95% confidence intervals shown in Figure 2. The frontier line increases with increasing clay + silt content to around 20%–45%, after which the rate of increase slows. The C<sub>Amax</sub> ranges from 4.8 g/kg soil to 45.66 g/kg soil with a mean of 32.4 g/kg soil (Table 1). The C<sub>def</sub> ranges from none to 45.05 g/kg soil with a mean of 25.95 g/kg soil (Table 1).

Table 1. Summary statistics

|                               | Mean  | SD    | Min  | $Q_{0.25}$ | Median | Q <sub>0.75</sub> | Max   | Skew  |
|-------------------------------|-------|-------|------|------------|--------|-------------------|-------|-------|
| Silt%                         | 10.93 | 7.48  | 0.54 | 4.73       | 9.49   | 16.31             | 31.81 | 0.63  |
| Clay%                         | 20.79 | 11.16 | 2.34 | 11.84      | 18.68  | 29.39             | 54.25 | 0.49  |
| MAOC (g/kg soil)              | 6.52  | 7.32  | 0.27 | 2.07       | 4.17   | 7.88              | 50.04 | 2.79  |
| C <sub>Amax</sub> (g/kg soil) | 32.76 | 10.52 | 5.29 | 26.84      | 36.15  | 41.24             | 45.80 | -0.89 |
| $C_{def}$ (g/kg soil)         | 26.31 | 11.22 | 0.00 | 19.15      | 28.59  | 35.65             | 45.17 | -0.64 |

Note: SD = Standard Deviation, Min = Minimum,  $Q_{0.25}$  = Lower 25% quartiles, Med = Median,  $Q_{0.75}$  = Upper 25% quartiles, Max = Maximum, Skew = Skewness.

Figure 2. Frontier lines and its 95% confidence interval fitted using all 482 samples.




# 3.2 Spectroscopic modelling of MAOC content

The CUBIST model predicts MAOC with an RMSE of 2.77 g/kg soil, is unbiased with R<sup>2</sup> of 0.86, and CCC of 0.91 Table 2, Figure 3 b). The model partitions the data into four rule sets, corresponding to different MAOC content levels, which increase from Rule 1 to Rule 4 (Figure 3a). Samples in Rule 1 have the least MAOC and are not significantly different from Rule 2 (Figure 3 a). Rule 3 samples have significantly more MAOC than Rule 1 but are not significantly different from Rule 2 (Figure 3 a). Rule 4 samples have significantly more MAOC than all other rules and exhibit the largest spread (Figure 3 a).

**Table 2.** Tuning parameters and model statistics for MAOC and  $C_{def}$  CUBIST models.

|                            | Committee | Neighbor | RMSE (g/kg soil) | $\mathbb{R}^2$ | CCC  |
|----------------------------|-----------|----------|------------------|----------------|------|
| MAOC                       | 1         | 8        | 2.77             | 0.86           | 0.91 |
| Mean $C_{def}$             | 1         | 5        | 3.72             | 0.89           | 0.94 |
| $C_{\it def}$ upper 95% CI | 1         | 4        | 4.13             | 0.85           | 0.92 |
| $C_{def}$ lower 95% CI     | 1         | 9        | 3.74             | 0.91           | 0.95 |

Note: RMSE = Root mean square error, CCC = Lin's concordance correlation coefficient, CI = Confidence interval.

The mean Mid-IR spectra of the samples of the four rule sets show overall consistent patterns, with differences in absorption intensities at 3700–3500 cm<sup>-1</sup>, 2946–2850 cm<sup>-1</sup>, 1986–1794 cm<sup>-1</sup>, and 1634–1300 cm<sup>-1</sup>) (Figure 3, c). Specifically, the mean spectrum of rule 4 has the highest absorption in the 2946–2850 cm<sup>-1</sup> region associated with organic C (C–H vibrations of Alkyl CH<sub>2</sub>), corresponding to having the highest MAOC content (Figure 3, a, c). The wavenumbers selected for each of the four rules' linear models differ, although there is some overlap. All rules use wavenumbers between 2946–2850 cm<sup>-1</sup>, organic C–H vibrations of Alkyl CH<sub>2</sub> groups (Nguyen et al., 1991) and near 2515 cm<sup>-1</sup> associated with carbonate (Nguyen et al., 1991) though the specific selections vary (Figure 3, c). Rule 1 exhibits densely distributed wavenumbers across both these regions with high coefficient values. Rule 3 shows a similarly dense distribution, concentrated primarily in the 2946–2850 cm<sup>-1</sup> region, with large coefficient values. In contrast, Rule 2 displays more sparsely distributed wavenumbers across both regions, while Rule 4 uses only a few select wavenumbers around 2946–2850 cm<sup>-1</sup>. Rules 1, 2, and 3 use the region between 1986–1794 cm<sup>-1</sup>, associated with quartz, with the coefficients in rule 2 having the largest magnitude (Figure 3, c). Rule 4 uniquely includes absorptions at the 3750 cm<sup>-1</sup> region, associated with the hydroxyl stretching vibrations of clay minerals (Nguyen et al., 1991); and between 1762–1634 cm<sup>-1</sup>, associated with amide C=O bond (Volkov et al., 2021), as well as wavenumbers around 1154 cm<sup>-1</sup>, which correspond to the SiO2 lattice (Spitzer and Kleinman, 1961) and C-OH stretch of aliphatic O–H (Senesi et al., 2003) (Figure 3, c).

**Figure 3.** CUBIST model result for MAOC. (a) The distribution of MAOC content for each CUBIST rule and Tukey's HSD between each CUBIST rule. (b) The correlation between observed and predicted MAOC of the CUBIST model, coloured by CUBIST rules. (c) The coefficient of each linear model for each CUBIST rule is plotted over the mean spectra of each CUBIST rule. (b) The correlation between observed and predicted MAOC of the CUBIST model coloured by CUBIST rules.





# 3.3 Spectral modelling of the organic C deficit $(C_{def})$

The model predicts  $C_{def}$  with an RMSE of 3.72 g/kg soil,  $R^2$  of 0.89, and CCC of 0.94 while also being unbiased (Table 2, 190 Figure 4 c). The model partitions the data into 3 rule sets, and the linear models of each CUBIST rule also show good precision (Table 3).

Rule 1 includes samples with the lowest  $C_{def}$  and the highest MAOC content, representing samples that have smaller C sequestration potential, as these samples contain more MAOC (Figure 4 a, b). Rule 2 represents samples with intermediate  $C_{def}$ , and contain little MAOC and clay and silt content, representing coarser-textured soils with more C sequestration potential than samples in rule 1 because they hold less MAOC (Figure 4 a, b). Rule 3 includes samples with high  $C_{def}$ , low MAOC content and the most clay and silt content. Since these samples contain the finest particles, their capacity is largest and is thus undersaturated with C relative to their potential (Figure 4 a, b).

**Table 3.** Model statistics for each linear model of the CUBIST rules in the mean  $C_{def}$  CUBIST model.

|        | RMSE (g/kg soil) | $\mathbb{R}^2$ | CCC  |
|--------|------------------|----------------|------|
| Rule 1 | 5.03             | 0.81           | 0.90 |
| Rule 2 | 2.25             | 0.94           | 0.97 |
| Rule 3 | 1.58             | 0.90           | 0.95 |

Note: RMSE = Root mean square error, CCC

The three rule sets show similar overall mean spectral patterns but with distinct differences in absorption intensities at key regions, including 2946–2850 cm<sup>-1</sup> associated with organic C, 1986–1794 cm<sup>-1</sup> associated with SiO<sub>2</sub> overtone and combination bands, and 1538–1218 cm<sup>-1</sup>) region associated with various organic and mineral absorptions (Figure 4 d). The wavenumbers selected for the models in each CUBIST rule are generally consistent, with the magnitude of the coefficient decreasing from rule 1 to rule 3 (Figure 4, d). In the 2946–2850 cm<sup>-1</sup> region, associated with organic C–H vibrations of Alkyl CH<sub>2</sub> groups (Nguyen et al., 1991), rule 1 has a greater average absorption compared to rule 2 and rule 3. This pattern corresponds to samples in rule 1 having the most MAOC content (Figure 4, b, d). All three CUBIST rules use wavenumbers within and near this region, and their coefficients are large. Rule 1 has the largest coefficients, followed by rules 2 and 3. Thus, rule 1 has the lowest  $C_{def}$  followed by rule 2 and 3 (Figure 4, b, d). The absorption near 2515 cm<sup>-1</sup> due to carbonates shows more prominent absorption in rule 3. In the region near 1986–1794 cm<sup>-1</sup>), which is due to the overtones of Si-O vibrations(Volkov et al., 2021), absorption intensity decreases from rule 2 to rule 1 to rule 3, corresponding to decreasing sand content and increasing clay and silt content (Figure 4, d). All three rules have prominent absorption at and near 1634 cm<sup>-1</sup>, which are associated with amide, carboxylate and carboxylic acid (Nguyen et al., 1991; Tanykova et al., 2021), aromatic -C=C-stretch (Du et al., 2014) HO–H stretch (Kronenberg et al., 1994), N–H bend, C=O stretch (Volkov et al., 2021) and absorbed water (Max and Chapados, 2009) (Figure 4, d). In the fingerprint region (1550–450 cm<sup>-1</sup>), the band assignments are more

<sup>=</sup> Lin's concordance correlation coefficient.

Figure 4. CUBIST model result for  $C_{def}$ , showing the CUBIST rules separation, including the distribution of (a)  $C_{def}$  and (b) MAOC content for each CUBIST rule and Tukey's HSD between each CUBIST rule. Along with (c) the correlation between observed and predicted  $C_{def}$  of the CUBIST model coloured by CUBIST rules, and (d) the coefficient of each linear model for each CUBIST rule plotted over the mean spectra of each CUBIST rule.



Figure 5. The correlation between observed and predicted  $C_{def}$  of the CUBIST model coloured by CUBIST rules, as well as the observed and predicted  $C_{def}$  estimated from the upper 95% CI and lower 95% CI of the frontier line fit.

challenging due to significant overlaps between mineral and organic absorptions (). The region from 1538 to 1218 cm<sup>-1</sup>, likely associated with quartz minerals as well as organic matter (Volkov et al., 2021), is more prominent in rule 2 and rule 1, and lower in rule 3 (Figure 4, d). Rule 3 exhibits proportionally larger coefficients for wavenumbers in the fingerprint region because of low organic C content and high fine mineral particle content (Figure 4, b, d).

The model statistics of the CUBIST models of  $C_{def}$  estimated from the upper and lower 95% CI of  $C_{Amax}$  are shown in Table 2. The model for the  $C_{def}$  estimated with the lower 95% CI of the frontier line performs better than the model estimated with the upper 95% CI. This can be attributed to the upper 95% CI of the frontier line having higher uncertainty than the lower 95% CI. Specifically, the upper uncertainty of the frontier line fit is high around 25% clay + silt content due to the low sample number (Figure 2). The uncertainty of  $C_{def}$  estimated from CUBIST models of  $C_{def}$  calculated from the upper CI and lower CI of the  $C_{Amax}$  is shown in Figure 5.

# 3.4 $C_{def}$ model interpretaion with SHAP

The SHAP contribution of spectral absorption at each wavenumber for the linear model of each CUBIST rule is shown in Figure 6. The SHAP values coincide with the regression coefficients of the CUBIST rules (Figure 6). The regression coefficients and




SHAP values are consistent, as large coefficients exhibit strong SHAP model contributions. Rule 1 shows strong contributions primarily from organic C features, and rule 2 displays a similar pattern but with more contributions from the fingerprint region. For rule 3, there is a relatively stronger contribution from the absorptions in the double bonds region (including absorption from quartz and the region associated with amide overlapping with other absorptions), and the fingerprint regions have a relatively stronger contribution (Figure 6).

The SHAP values indicate positive and negative contributions from spectral regions associated with characteristic absorption of clay minerals, organic matter, and quartz (Figure 6). Generally, peaks associated with organic C have a negative model contribution with an increase in absorbance, while the troughs have a positive contribution with increasing absorbance (Figure 6). Similarly, absorptions associated with clay minerals, quartz and silicate have a positive model contribution, while the troughs have a negative contribution (Figure 6).

#### 4 Discussion

Our findings support the hypothesis that mid-IR spectra, combined with machine learning and enhanced by SHAP analysis for interpretability, can accurately estimate soil MAOC content 2) and  $C_{def}$  (Table 2) by elucidating the contribution of specific mid-IR absorptions.

Our results demonstrate that combining soil spectroscopy with machine learning offers a rapid, cost-effective, and robust method for estimating MAOC and C<sub>def</sub>. The spectroscopic approach enables many more measurements than conventional methods, enhancing our understanding of how MAOC and C<sub>def</sub> vary in the soil in space and time (Angers et al., 2011). This approach could also provide essential data for soil biogeochemical and Earth System models, improving their initialisation, validation and ongoing development (Stewart et al., 2007; Georgiou et al., 2022; Abramoff et al., 2022; Vereecken et al., 2016).

Given that C storage is a key soil function for maintaining soil health (Lal, 2016; Lehmann et al., 2020), our findings highlight how the current state and potential for C sequestration can be rapidly and cost-effectively measured as part of soil health assessment (Vogel et al., 2019). This aligns with growing evidence that soil spectra, when combined with machine learning, can model soil functions, going beyond the prediction of individual soil properties (Cohen et al., 2006; Elliott et al., 2007; Cécillon et al., 2009; Viscarra Rossel et al., 2010; Maynard and Johnson, 2018; Deiss et al., 2023).

Two other studies estimated soil  $C_{def}$  using mid-IR spectroscopic modelling Karunaratne et al. (2024); Baldock et al. (2019). Unlike these studies, which used quantile regressions to estimate  $C_{def}$ , our approach avoids under- or over-estimations (of  $C_{def}$ ) using bootstrapped frontier lines that more accurately capture the relationship between MAOC and clay + silt content (Viscarra Rossel et al., 2024a). Additionally, the spectroscopic  $C_{def}$  model was developed using CUBIST, which offers good predictability and interpretability, effectively handling non-linearities, and is advantageous compared to linear methods like PLSR. Unlike the earlier studies, we also propagated the uncertainties from the frontier lines fits and the CUBIST models to our final predictions.

The MAOC and  $C_{def}$  models relied on spectral regions related to organic functional groups such as the C-H groups near 2900 and 2800cm<sup>-1</sup>, the amide group near 1725cm<sup>-1</sup> and 1:1 and 2:1 clay minerals, which provide surfaces for organic matter

**Figure 6.** The mean spectra, key spectral assignment, and the SHAP contribution of the spectral regions used in each linear model of each CUBIST rule. A positive SHAP value indicates a positive contribution to a model with increased absorbance, whereas a negative SHAP value indicates a negative contribution with increased absorbance. The magnitude of SHAP indicates the strength of the contribution. The SHAP values are plotted over the pre-processed spectra of each rule set. The SHAP values are coloured by the normalised absorbance value at each wavenumber, ranging from -1 (lowest absorbance at each wavenumber) to 1 (highest absorbance at each wavenumber).








adsorption. Absorptions for quartz and other minerals in the fingerprint region were also important in the models, but negatively affected the estimates. The  $C_{def}$  model drew from information on C already present in the soil, which contributed negatively to the model, and soil mineralogy, which provides information on what the soil minerals could potentially adsorb, contributing positively to the model.

CUBIST offers an advantage over other machine learning models due to its interpretability. As a tree-based algorithm, it can be locally interpreted, unlike other algorithms that are limited to global-level interpretation (Viscarra Rossel and Webster, 2012). SHAP values provided additional interpretation, allowing us to not only know how each wavelength contributes to the model and how strongly they contributed to it but also show what direction an increase or decrease in absorbance affects the model, thus identifying which and how soil constituents (clay minerals, quartz, and organic C) significantly contribute to determining MAOC and  $C_{def}$  (Wadoux, 2023). Nevertheless, given the heterogeneous nature of soil composition, overlapping absorptions make it challenging to differentiate between molecular vibrations, particularly in the fingerprint region. Like other regression tree methods, CUBIST can be sensitive to strong collinearity, potentially leading to model instability and overfitting (Viscarra Rossel and Webster, 2012; Kuhn, 2013). To minimise the effect of collinearity in our modelling, we interpolated the spectra to a resolution of 32 cm $^{-1}$  (see Methods section).

This study extends beyond previous research, which predominantly focuses on agricultural land use, by incorporating samples from various other ecosystems. The samples span Australia's main Köppen-Geiger climate zones, 24 major vegetation groups, and 11 of the 14 Australian soil classification orders (Isbell et al., 2016). We excluded hydrosol with different C stabilisation dynamics. Future work will include more samples with a larger representation of soils for developing site-specific  $C_{def}$  estimates.

Our method facilitates efficient data acquisition, providing an effective approach to help farmers and land managers gain the insights needed to assess the current and potential for carbon sequestration on their land. Identifying regions and soil types where increasing organic C storage is feasible enables more targeted resource allocation and informed decision-making.

While our study pertains to Australian soils, the principles of applying mid-IR spectroscopy and machine learning to estimate MAOC and  $C_{def}$  are applicable across various land uses, soil types, and climatic conditions. This provides the rapid assessment capability needed to scale soil carbon initiatives for monitoring soil organic carbon and its potential contribution to climate adaptation and mitigation targets under the Paris Agreement and the UN Sustainable Development Goals. The method's ability to support large-scale monitoring of C sequestration potential also makes it relevant to soil carbon credit systems such as the Australian Carbon Credit Units (ACCU) scheme.

### 5 Conclusions

We demonstrated that mid-IR spectroscopy combined with machine learning could effectively estimate soil MAOC content (RMSE = 2.77 g/kg soil,  $R^2 = 0.86$ , CCC = 0.91) and  $C_{def}$  (RMSE = 3.72 g/kg soil,  $R^2 = 0.89$ , CCC = 0.94). We interpreted CUBIST, confirming the contributions to the models from functional groups related to organic functional groups, clay minerals, and quartz, reflecting existing soil organic C, soil mineralogy, particle size distribution, and surface area available for C

adsorption, which are critical for estimating MAOC and  $C_{def}$ . Our approach contributes to understanding the analysis of C sequestration potential with mid-IR spectroscopy and machine learning, helping the development of rapid and cost-effective soil C sequestration assessment and monitoring.

Author contributions. YH: Investigation, methodology, analysis, visualisation and writing. RAVR: Conceptualisation, methodology, writing, editing, supervision and funding acquisition.

Code and data availability. The code and dataset will be made available upon reasonable request.

Competing interests. At least one of the (co-)authors is a member of the editorial board of SOIL.

Acknowledgements. RAVR thanks the Australian Government's Australia-China Science and Research Fund-Joint Research Centres (ACSRF-300 JRCs) (grant ACSRIV000077) and the Australian Research Council's Discovery Projects scheme (project DP210100420) for funding. We also thank Mr. Farid Sepanta for the laboratory analyses of the soils, and Drs. Zefang Shen and Adam Cross for earlier project discussions. We are grateful to the Terrestrial Ecosystem Research Network (TERN) and Dr Andrew Bissett, who provided us with some of the soil samples used in the work.

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
