# Peer review of "Estimating soil carbon sequestration potential with mid-IR spectroscopy and explainable machine learning"

_EGUsphere, 2025_

## Author Comment (AC1)

**Response to referee 1**

We thank the editor and referee for their comments and reviews. Below, we respond to each comment (in blue text and preceded by **Authors:**).

**Referee**

**Comment** General comments: Based on national scale soil samplings, this manuscript proved the potential of implementing mid-IR spectra and machine-learning for MAOC and C deficit prediction. The results show that the CUBIST models for both MAOC and C deficit prediction have good performance, advocating their future application. They also make these models interpretable by matching absorption features of the mid-IR spectra and coefficients in models among different modeling rules. Nevertheless, several issues raised during my review which I think should be addressed before publication.

The investigation of model interpretability should be modified. Since the SHAP values coincide with the regression coefficients of the CUBIST rules, there is large redundancy between the SHAP analysis and that of CUBIST rules demonstration. In other words, the interpretation that positive SHAP values had a positive impact on the model prediction also applies to that of coefficient values in multivariate regression. The authors should demonstrate the additive value of the SHAP analysis. In addition, if the authors manage to do so, then they should also perform the SHAP analysis on MAOC prediction model. Otherwise, the authors should declare the reason why they only perform the SHAP analysis on C deficit prediction model. In addition, the so-called interpretability stops by pointing out impactful wavenumber and its chemical identity. The interpretability should involve more explanatory descriptions. For instances, in line 259, "absorptions for quartz and other minerals in the fingerprint region were also important in the models, but negatively affected the estimates". What did this result tell us? Is that because the relatively larger amount of quartz likely indicates a sandy texture of soils, thus indicating less mineral

capacity and likely low C deficit?

**Authors:** We thank the reviewer for these comments on model interpretability. We agree that, in linear models, signed coefficients and SHAP values can convey similar information about the average direction of effect. However, in our CUBIST implementation, regression coefficients are defined by rules that apply to subsets of the data, whereas SHAP values quantify each wavenumber's contribution to the prediction for each individual sample. In our context, the two diagnostics are therefore not redundant. Regression coefficients summarise the average effect of a wavenumber within a given rule, while SHAP values provide instance–level attributions that reveal how the same wavenumber can contribute differently across samples depending on their full spectral and covariate context [Shapley, 1953]. This is illustrated in Figure 6, where the magnitude and spread of SHAP contributions for key wavenumbers differ from those inferred from the rule coefficients alone, highlighting sample-specific effects that are not apparent from the rule-based coefficients.

Regarding the scope of the SHAP analysis, our primary objective in this study is to model and interpret the carbon deficit (Cdef), with the MAOC model serving as a supporting component in deriving $C_{def}$. For this reason, we focused the main-text SHAP analysis on the $C_{def}$ model. In response to the reviewer's suggestion, we will now include an analogous SHAP analysis for the MAOC prediction model in the Supplementary Information, and cross-reference it in the main text, so that readers can assess whether influential wavenumbers and their signs are consistent across both models.

We agree that interpretability should extend beyond listing important wavenumbers to providing ecological and pedological explanations. We thought we had done that, but we will revise thoroughly to ensure we do! The negative SHAP values associated with quartz-related absorptions in the fingerprint region indicate that, all else being equal, spectra dominated by quartz and other primary minerals

tend to be associated with lower predicted $C_{def}$. This is consistent with the interpretation that higher quartz content reflects coarser-textured, sandier soils with lower specific surface area and reduced capacity to stabilise mineral-associated organic carbon. Conversely, lower quartz and stronger absorptions linked to clay minerals and organo–mineral complexes are associated with higher predicted $C_{def}$, in line with greater mineral capacity for sorption and stabilisation. We will clarify this interpretation in the revised text, so that the SHAP patterns are explicitly linked to soil texture, mineral surface area, and mineral-associated C storage capacity.

**Comment** The discussion section should be modified in several aspects. First, the authors stated that the spectroscopic approach enables many more measurements than conventional methods, enhancing our understanding of how MAOC and C deficit vary in the soil in space and time. However, the approach that this study implemented still involved destructive samplings over large geographical scale, which still belong to conventional methods. In other words, in order to monitor C deficit dynamics over time, researchers need long-term large-scale samplings to get the new mid-IR spectra from soils, even they have built the CUBIST models. Therefore, the statement will be a better fit for spectroscopic approaches which use spectra from non-destructive remote sensing techniques, i.e. spectra from satellites, even though the model accuracy of these studies tends to be lower than this study. If insist using the statement mentioned above, the authors should point out the potential that laboratory-based spectroscopic approaches can help improve the performance of that of remote sensing spectroscopic approaches. Second, the authors pointed out that the frontier line approach can have a more accurate estimate of MAOC maximum capacity than that of quantile regression in discussion part. However, Shi et al (doi.org/10.1016/j.geoderma.2025.117181) has implemented a local approach for the quantile regression method, which has the merit of avoiding under- or over-estimations. The authors should incorporate Shi's study into the discussion section and modify the relevant statements.

**Authors:** We thank the reviewer for the comments on the Discussion. We agree that the mid-IR spectroscopy implemented in this study still relies on destructive soil sampling and laboratory measurements, and therefore differs fundamentally from non-destructive satellite-based remote sensing. Our intention was not to imply that mid-IR spectroscopy eliminates the need for sampling, but rather to emphasise that, once a representative calibration set has been analysed, laboratory mid-IR spectroscopy allows MAOC and related C fractions to be estimated for far larger numbers of samples than would be feasible with conventional wet-chemical fractionation and combustion methods, which are labour-intensive, time-consuming, and costly. We will clarify this point in the revised Discussion.

We also acknowledge that satellite-based spectroscopic approaches can provide truly non-destructive, repeated observations at very large spatial scales. However, optical remote sensing is restricted to the soil surface (millimtetres), is strongly affected by vegetation cover, soil moisture, and atmospheric variability, and lacks the spectral resolution and sensitivity of laboratory mid-IR spectroscopy for resolving the mineral and organic functional groups that govern MAOC and $C_{def}$. Moreover, any satellite-based estimation of $C_{def}$ would require extensive ground truthing and calibration against laboratory measurements. To avoid over-stating our contribution, we will soften the wording in the Discussion and explicitly state that our laboratory-based spectroscopic framework is primarily intended to support high-throughput MAOC and $C_{def}$ estimation on sampled soils, and that such laboratory models can in future underpin and improve the calibration and validation of remote-sensing-based approaches.

We thank the reviewer for pointing us to the work of Shi et al. (2025). Their study reduces under- and over-estimation of carbon sequestration potential by using local quantile regressions on spectrally defined, mineralogically similar subsets of the data, rather than a single global quantile relationship. This local approach is an important advance that better respects spatial and mineralogical heterogeneity. At

the same time, quantile regression, whether global or local, still fits a parametric relationship through the upper portion of the MAOC–(clay + silt) distribution, so that some observed values can lie above the fitted line and the implied MAOC capacity continues to increase linearly with increasing fine fraction

In contrast, the frontier-line approach we adopt explicitly estimates the upper envelope of the MAOC–(clay + silt) relationship under current environmental conditions and quantifies the associated uncertainty. This upper-envelope formulation inherently prevents observed values from lying above the estimated capacity line and allows the frontier to level off at high clay + silt contents, reflecting finite organic inputs and diminishing stabilisation efficiency, for example where rainfall limits plant production. As a result, frontier-line analysis reduces both underestimation (by targeting the effective upper boundary rather than an internal quantile) and overestimation (by avoiding unconstrained extrapolation at high clay + silt), providing more realistic estimates of attainable MAOC storage under prevailing climate and mineralogical constraints. We will revise the Discussion to (i) acknowledge the contribution of local quantile regression approaches such as Shi et al. (2025), (ii) clarify how the frontier-line method relates to and extends these efforts, and (iii) focus on the specific advantages of the frontier-line formulation for estimating effective MAOC capacity.

**Comment** The particle size of clay and silt content and of fine fraction in soil fractionation are methodologically mismatched, which induced errors. The mineral capacity between soil particles under 20 $\mu$m and 50 $\mu$m are different. Because these two sets of soil minerals have different structure in their components. For instance, the 50 $\mu$m set might constitute more quartz, feldspar, and 1:1 type clay mineral, which have lower C absorption capacity than that of 2:1 type clay mineral. Thus, the C absorption capacity of soil minerals partitioned by 20 or 50 micrometers cannot represent each other. Using 20 $\mu$m clay and silt content to capture MAOC maximum capacity corresponding to 50 $\mu$m fractionation protocols does not robustly reflect the

relationship. There might be a few options for improvement: changing the model for clay and silt prediction, laboratory work for clay and silt content, or at least acknowledging this limitation in the discussion.

**Authors:** We thank the reviewer for raising this methodological point. We measured MAOC as soil organic C in the $\leq 50$ $\mu$m fraction, whereas clay and silt contents were defined using the Australian particle-size classification (clay ¡2 $\mu$m, silt 2–20 $\mu$m, fine sand 20–200 $\mu$m). We agree that the $\leq 20$ $\mu$m clay + silt fraction is not mineralogically equivalent to the $\leq 50$ $\mu$m fraction, and that this mismatch can introduce error into the empirical relationship between MAOC and fine fraction because the 20–50 $\mu$m size range can contain additional primary minerals (e.g. quartz, feldspar) and 1:1 clays with lower C sorption capacity than finer 2:1 clays and oxides.

Both the $\leq 50$ $\mu$m MAOC fraction and the 2–20 $\mu$m particle-size classes follow established, standardised methods for soil fractionation and texture analysis used in Australian and international studies, and our approach is consistent with recent national-scale work on soil C fractions [Walden et al., 2025].

We considered alternative options, including re-estimating fine fractions directly from laboratory particle-size analyses and developing new models specifically targeting a $\leq 50$ $\mu$m definition. However, for this national-scale dataset, the available particle-size information is most consistently reported according to the ¡2 and 2–20 $\mu$m classes, and re-fractionation of all samples was not feasible within the scope of the present study. We therefore used $\leq 20$ $\mu$m as an operational proxy for the mineral-associated fraction, while recognising that it does not fully capture the mineralogical composition of the $\leq 50$ $\mu$m MAOC fraction.

Because many Australian soils have relatively low silt contents and a large proportion of the fine fraction is in the clay size range, the practical impact of the 20–50 $\mu$mm gap on estimated mineral surface area may be smaller in some regions than in silt-rich soils elsewhere; however, without direct measurements of the 20–50 $\mu$mm sub-fraction, we cannot quantify this effect.

In the revised Discussion, we will explicitly acknowledge this methodological inconsistency and note that it may contribute to both scatter and potential bias in our estimated MAOC–(clay + silt) relationships. We will also emphasise that future work should, where possible, align the operational definitions of fine fractions used for MAOC measurement and for particle-size characterisation (e.g. by directly measuring and modelling $\leq 50$ $\mu$m or $\leq 53$ $\mu$m silt + clay fractions) to better reflect the true mineral capacity for C stabilisation.

**Minor comments**:

**Comment** Line 41: Instead of fitting 90th quantile regression, Georgiou et al used 95th quantile regression. Please check.

**Authors:** Thanks they indeed used 95th quantile. We will correct this in the manuscript.

**Comment** Line 116: Did this back-transformation be performed during uncertainty analysis? Since the authors used logarithm when fitting the frontier line, the upper and lower uncertainty intervals would be different between that undergone first calculating intervals then back-transformation, and that undergone first back-transformation then calculating intervals. Please clarify.

**Authors:** In line 116 and 117 we did specify the back transformation was performed after the uncertainty calculation. We will make this more obvious to the reader in the revision.

**Comment** Line 124: What specific are the offset corrections? SNV transformation is well-known in spectroscopic area, while offset correction tend to be a series of mathematical operation on the spectra. Please clarify or at least provide reference.

**Authors:** The offset correction we applied here was simply subtracting its smallest value (minus 0.01) from all its measurements, so every spectrum's lowest point aligns just above zero, which removes background bias caused by ambient light or imperfect calibration, making all spectra comparable on the same baseline. We will clarify this in the method section.

**Comment** Line 174-176: The result is not intuitive. It is hard to tell whether samples in Rule 3 have higher absorption in the 2946-2850 cm-1 region than that of Rule 4, given the scale of the y-axis in the two plots are not consistent. Could the authors please make this comparison more intuitive, thus better supporting the statement?

**Authors:** Thank you for pointing this out. We will revise Figure 3 to make sure the spectra for each rules are plotted on the same scale.

**Comment** Line 255: The authors mentioned they have propagated the uncertainties from the frontier lines fits and the CUBIST models to our final predictions. Do the uncertainties of the frontier line fits have anything to do with the uncertainty of C deficit CUBIST model? Because the latter is demonstrated with parameters like RMSE only for C deficit model not its upper or lower 95% confidence intervals CUBIST models. There is a mismatch between the grey areas in Figure 5 and statistical parameters of the C deficit CUBIST model, indicating there is no propagation of the intervals to the final C deficit prediction. Please clarify.

We thank the reviewer for highlighting this point about uncertainty propagation. Our intention was not to imply that we combined the frontier-line and CUBIST uncertainties into a single scalar summary (e.g. a single RMSE that incorporates both), but rather that we quantified and presented uncertainty from both modelling steps that contribute to $C_{\text{def}}$.

Specifically, there are two sources of uncertainty in our framework: (1) the non-parametric frontier-line fit of MAOC against clay + silt, for which we estimate upper and lower 95% confidence limits, and (2) the CUBIST model used to predict Cdef from mid-IR spectra, for which we report standard performance metrics, including RMSE, based on cross-validation. Both sources affect the final Cdef estimates because the frontier line defines the target Cdef values that CUBIST learns, and variation of the frontier within its confidence limits leads to different Cdef targets and hence different predictions.

The grey envelopes in Figure 5 do not represent the RMSE of the $C_{def}$ CUBIST model, but the range of $C_{def}$ predictions obtained when CUBIST is applied to the upper and lower 95% confidence limits of the frontier-line fit. In other words, the grey area reflects the propagated effect of uncertainty in the frontier-line estimates on the $C_{def}$ predictions, whereas the RMSE reported for the $C_{def}$ model quantifies the predictive error for the mean (best-estimate) $C_{def}$ values. We will clarify this explicitly in the revised text and figure caption to avoid any ambiguity.

We acknowledge that a fully joint treatment of uncertainty (e.g. providing formal prediction intervals that simultaneously incorporate both frontier-line and CUBIST model uncertainty for each prediction) would require more complex statistical analysis and is beyond the scope of this initial study. Nevertheless, we believe that presenting (i) confidence limits for the frontier-line capacity estimates and (ii) cross-validated performance metrics for the $C_{def}$ CUBIST model, together with the grey envelopes showing how frontier-line uncertainty propagates to $C_{def}$ predictions, provides a transparent account of the main uncertainty sources in our approach.

**References**

Lloyd S Shapley. A value for n-person games. Contribution to the Theory of Games, 2, 1953.

Lewis Walden, Farid Sepanta, and RA Viscarra Rossel. Ft-mir spectroscopic analysis of the organic carbon fractions in australian mineral soils. European Journal of Soil Science, 76(2):e70084, 2025.